# Knowledge, attitudes and factors associated with the awareness of caregivers of under-five children regarding the malaria vaccine in the Tiko Health District, Cameroon: A community-based cross-sectional study

**Maureen Abiache Idang[1], Divine Nsobinenyui[2], Chrisantus Eweh Ukah[3]\*, Larissa Kumenyuy Yunika[3], Claudia Ngeha Ngu[3], Randolf Fuanghene Wefuan[3], Syveline Zuh Dang[3], Ndip Esther Ndip[3], Mirabelle Pandong Feguem[3], Claudine Mulih Shei[3], Dickson S. Nsagha[3]**

**1** Faculty of Medicine and Pharmaceutical Sciences, Saint Monica University Higher Institute of Buea, Buea, Cameroon, **2** Department of Applied Zoology, Faculty of Science, University of Bamenda, Bamenda, Cameroon, **3** Department of Public Health and Hygiene, Faculty of Health Sciences, University of Buea, Buea, Cameroon

\* chrisantuseweh@gmail.com

## Abstract

Despite progress in malaria control, malaria remains a major public health burden in sub-Saharan Africa, particularly among children under five. The introduction of malaria vaccines, including RTS,S/AS01 (Mosquirix) and the recently WHO-recommended R21/Matrix-M, offers renewed hope for reducing malaria morbidity and mortality. The effectiveness of these vaccines, however, depends largely on caregivers' awareness, knowledge, and attitudes. This study assessed caregivers' knowledge and attitudes, and the factors associated with awareness of the malaria vaccine in the Tiko Health District of Cameroon. A community-based cross-sectional study was conducted among 410 caregivers of children aged 0–5 years who were selected using a multistage sampling technique. Data were collected using a structured pre-tested questionnaire. Descriptive statistics summarized participants' characteristics, and knowledge and attitude scores were generated using a structured scoring system with a 60% cut-off defining adequate knowledge and positive attitudes. Logistic regression analysis identified factors independently associated with malaria vaccine awareness with statistical significance set at p < 0.05. The median age of participants was 32 years(IQR:27–40), and most were female(83.2%). Although 60.7% of caregivers had heard of the malaria vaccine, only 26.6% demonstrated adequate knowledge and 25.1% had positive attitudes. Healthcare workers were the primary source of vaccine information(35.4%). Caregivers whose children had a previous malaria episode were less likely to be aware of the vaccine(AOR:0.55; 95% CI:0.28–0.97). Conversely, caregivers who trusted health workers (AOR:3.02; 95% CI:1.83–4.99)

**Data availability statement:** All relevant data are within the paper and its Supporting Information files. The de-identified dataset underlying the findings of this study has been provided as a Supporting Information file in Excel format.

**Funding:** The authors received no specific funding for this work.

**Competing interests:** The authors have declared that no competing interests exist.

and those who routinely attended childhood immunization services (AOR:3.57; 95% CI:2.27–5.60) were more likely to be aware of the vaccine. Caregivers in the Tiko Health District exhibited limited knowledge and generally negative attitudes toward the malaria vaccine. Strengthening health-worker engagement, improving communication during routine immunization services, and addressing gaps in caregivers' understanding may enhance malaria vaccine uptake in the district.

## Introduction

Malaria remains one of the most significant public health challenges globally, particularly in sub-Saharan Africa where it disproportionately affects vulnerable populations, including children under five years of age [1–4]. According to the World Health Organization (WHO), malaria accounted for an estimated 619,000 deaths in 2021, with young children representing a substantial proportion of these fatalities [5]. Despite ongoing efforts to control malaria through preventive measures such as insecticide-treated nets and antimalarial medications, the disease continues to pose a severe threat to child health in endemic regions [6,7].

The development of malaria vaccines has emerged as a promising strategy to complement existing prevention and treatment measures [8]. The RTS,S/AS01 (Mosquirix) vaccine, the first malaria vaccine to receive a WHO recommendation for use in children has shown efficacy in reducing malaria incidence and severe disease in clinical trials [9–12]. In Cameroon, the RTS,S/AS01 malaria vaccine was officially approved for introduction into the national Expanded Program on Immunization (EPI) in 2023 following the World Health Organization recommendation for its broader use [13]. Implementation of the malaria vaccine in Cameroon began in selected pilot health districts in late 2023, with gradual scale-up to additional districts in 2024 [14]. In the Tiko Health District, the malaria vaccine became available to eligible children through routine immunization services on January 22, 2024, prior to the conduct of this study [14]. This timing is important for interpreting caregivers' awareness and knowledge, as exposure to malaria vaccine–related information in the district was still at an early phase during the period of data collection. However, the successful implementation of vaccination programs is contingent upon the awareness, knowledge, and attitudes of caregivers, who play a crucial role in the health-seeking behaviors of their children [15–17].

In the Tiko health district, where malaria transmission is high understanding caregivers' perceptions and knowledge about the malaria vaccine is essential for developing effective public health strategies [18]. Caregivers' attitudes towards vaccination can significantly influence vaccine uptake and adherence, ultimately affecting the overall impact of vaccination campaigns. Factors such as cultural beliefs, previous experiences with healthcare services, and access to information can shape these attitudes and knowledge levels [19,20].

Research has indicated that misinformation and lack of awareness about vaccines can hinder vaccination efforts leading to lower coverage rates [21,22]. Moreover,

caregivers' socio-economic status, education level, and exposure to health education initiatives can further affect their understanding and acceptance of new interventions like the malaria vaccine [23]. Therefore, assessing caregivers' knowledge and attitudes towards the malaria vaccine in the Tiko Health District is critical for identifying barriers to vaccination and informing tailored educational interventions.

This study aims to fill the gap in knowledge regarding caregivers' perspectives on the malaria vaccine for under-five children in the Tiko health district. By exploring these dimensions, we hope to contribute valuable insights that can enhance vaccination strategies and ultimately reduce the burden of malaria among young children in this high-risk area.

This study was guided by the Health Belief Model (HBM), a widely used theoretical framework for understanding health-related decision-making and preventive behaviors. The HBM posits that individuals' engagement in health actions is influenced by their perceived susceptibility to a disease, perceived severity of its consequences, perceived benefits of an intervention, perceived barriers to action, cues to action, and self-efficacy [24].

In the context of malaria vaccination, caregivers' awareness, knowledge, and attitudes may be shaped by their perceptions of malaria risk to their children, beliefs about the effectiveness and safety of the vaccine, trust in health workers as cues to action, and prior experiences with malaria and routine immunization services [25,26]. By applying the HBM, this study conceptualizes caregivers' awareness of the malaria vaccine as a function of both cognitive (knowledge and beliefs) and contextual factors (health system trust and vaccination practices).

## Materials and methods

### Research design

This study used a cross-sectional study design to assess the knowledge and attitudes of caregivers of children age 0-5 years in the Tiko Health District

### Study area

The study was carried out in the Tiko Health District of the Southwest Region of Cameroon from 1st of March 2025 to the 11th of April 2025. Tiko Health District (THD), found in Fako Division, South West Region of Cameroon is located between latitude 9˚32'2"N to 9˚40'9"N and longitude 9˚25'7"E to 9˚55'7"E and an altitude of 32.84 m (107.76 ft). Tiko District, located in the Southwest Region of Cameroon, is divided into eight health areas. These health areas are managed by the local health administration and serve as the primary zones for delivering healthcare services to the community. Each health area typically includes a number of health centers and posts that provide essential services, such as maternal and childcare, immunization, treatment for infectious diseases like malaria, and other public health services.

### Study settings and duration

This study was conducted from 1 March 2025 to 11 April 2025 in selected health areas of the Tiko Health District.. The six of the eight selected health areas were: Likomba, Holforth, Mutengene, Misselele, Tiko Town, and Kange.

### Target topulation

The target population for this study was caregivers of children 0–5 years old resident within the Tiko Health District.

### Sample size

The sample size was obtained using the formula for estimation of confidence interval for a proportion.

$$n = \frac{z^2 * P(1-p)}{e^2}$$

Where;

 Z= 1.96 (for 95% confidence level)

 P= 0.5 (estimated proportion of caregivers' awareness),

 e= 0.05 (margin of error)

$n = \frac{1.96^2 * P(1-0.5)}{0.05^2} = 384$. To compensate for possible non-response and enhance the precision of estimates, the minimum sample size of 384 was increased to 410 participants.

## Sampling technique

This study employed a multi-stage sampling technique to assess the knowledge, attitudes, and factors associated with under-five awareness regarding the malaria vaccine within the Tiko health districts. The Tiko health districts comprise eight distinct health areas. To ensure a representative sample while maintaining feasibility, six of these health areas were randomly selected by balloting. This approach allowed for an unbiased selection process, ensuring that each health area had an equal opportunity to be included in the study. Following the selection of the health areas, community health workers were engaged to identify households with children under the age of five. These health workers possess intimate knowledge of their respective communities and were instrumental in locating eligible households. The identification process involved collaboration with community health workers who provided a list of households with under-five children within the selected health areas. A probability proportionate to size sampling was then used to determine the number of caregivers to be selected from each of the selected six health areas (Table 1)

Once the households were identified, caregivers of under-five children were approached for participation in the study. The selection of participants was based on their availability and willingness to participate, ensuring that informed consent was obtained prior to data collection. This systematic approach facilitated the recruitment of a diverse sample of caregivers, thereby enhancing the reliability and validity of the findings regarding their knowledge and attitudes, factors associated with their awareness regarding the malaria vaccine. The sampling technique utilized in this study involved a combination of random selection of health areas and purposeful identification of households by community health workers, which collectively contributed to a robust methodology for understanding caregiver perspectives in the Tiko health districts. Caregivers in identified households were recruited consecutively until the until the determined sample size per health area was reached.

## Inclusion criteria

Caregivers of under-five children living in the Tiko health district who gave their consent were included in the study.

## Exclusion criteria

Under-five children's caregivers who were severely ill at the time of data collection.

**Table 1. Number of caregivers selected per selected health area.**

| Health Area | Estimated caregivers Population | Proportion of caregivers | Allocated Sample (n) |
|---|---|---|---|
| Holforth | 5,706 | 0.28 | 113 |
| Mutengene | 8,170 | 0.40 | 162 |
| Likomba | 1,798 | 0.09 | 36 |
| Misselele | 955 | 0.05 | 19 |
| Kange | 709 | 0.03 | 14 |
| Tiko Town | 3,303 | 0.16 | 66 |
| **Total** | **20,641** | **1.00** | **410** |

## Data collection tools and methods

Data collection for this study was conducted using a pre-tested structured questionnaire designed to gather comprehensive information on the knowledge and attitudes of caregivers of under-five children toward the malaria vaccine. The questionnaire was pre-test among 20 caregivers in the Buea Health District which was not part of the study area. The questionnaire was systematically divided into three distinct sections to facilitate focused data collection:

### Section A: Socio-demographic variables

This section collected essential demographic information, including age, level of education, religion, location, and marital status of the caregivers. These variables provide context for understanding the background of participants and their potential influence on knowledge and attitudes toward the malaria vaccine.

### Section B: Knowledge of caregivers

This section assessed the knowledge of caregivers regarding the malaria vaccine through ten closed-ended questions. These questions were designed to evaluate caregivers' understanding of malaria, its transmission, prevention strategies, and specific details about the malaria vaccine.

### Section C: Attitudes toward the malaria vaccine

This section focused on the attitudes of caregivers toward the malaria vaccine, comprising ten closed-ended questions. The questions aimed to capture caregivers' perceptions, beliefs, and feelings about the vaccine, including any concerns or misconceptions they may have.

To ensure the validity and reliability of the questionnaire, a pre-test was conducted among ten under-five children in the Buea Health District at the Tole Health Area. This pre-test allowed for adjustments to be made to improve clarity and applicability during data collection.

### Conceptual framework: Health Belief Model (HBM)

The study was guided by the Health Belief Model (HBM), which is commonly used to explain and predict health-related behaviors, including vaccine acceptance. The HBM posits that individuals' decisions to adopt a health behavior are influenced by six key constructs:

1. **Perceived susceptibility** – caregivers' beliefs about their child's likelihood of contracting malaria.
2. **Perceived severity** – beliefs regarding the seriousness of malaria and its consequences.
3. **Perceived benefits** – caregivers' beliefs about the effectiveness of the malaria vaccine in preventing malaria.
4. **Perceived barriers** – concerns about vaccine safety, side effects, or misinformation that may prevent vaccination.
5. **Cues to action** – triggers such as advice from health workers, community messages, or prior child illness that motivate caregivers to seek vaccination.
6. **Self-efficacy** – caregivers' confidence in their ability to access vaccination services for their children.

The questionnaire was developed to incorporate these constructs, and variables included in the analysis were mapped onto the HBM components. The model also informed the interpretation of factors associated with caregivers' awareness of the malaria vaccine.

## Data collection

Data collection was carried out by the principal investigator and four trained research assistants using the structured questionnaires. The process involved both self-administration for caregivers who were literate and interviewer administration for those who required assistance. For literate participants, the questionnaires were self-administered; for those who were unable to read or write, the research assistants read the questions aloud and recorded their responses.

Prior to participation, all caregivers were adequately informed about the study through a written information sheet and detailed verbal explanations. Written informed consent was obtained from each participant before proceeding with data collection. Participants were made aware of their rights to withdraw from the study at any time without any repercussions. Confidentiality was strictly maintained by anonymizing responses; no names were recorded on the questionnaires. Instead, each questionnaire was assigned a unique file number, accessible only to the investigator for data analysis purposes.

The research assistants underwent a comprehensive training program that included a detailed training agenda and manual. Training covered essential topics such as data collection techniques, community engagement strategies, ethical considerations, and maintaining confidentiality throughout the study process. This training ensured that all research assistants were well-prepared to conduct data collection effectively and ethically.

## Ethical considerations

Ethical clearance was obtained from the Ethics Committee for Human Health Research in the Southwest Region of Buea. Additionally, administrative authorizations were secured from the Faculty of Health Sciences Institutional Review Board at the University of Douala, which were subsequently submitted to the Tiko District Health Services prior to data collection.

Informed consent was obtained from all participants prior to their inclusion in the study. Participants were provided with a detailed consent form that outlined the purpose of the research, the procedures involved, potential risks and benefits, and their right to withdraw from the study at any time without any consequences. This ensured that participants had a clear understanding of their involvement and could make an informed decision regarding their participation.

Confidentiality was strictly maintained throughout the research process. All data collected were anonymized, and identifying information was removed to protect participants' privacy. Data were stored securely and accessible only to authorized research personnel. Findings were reported in aggregate form, ensuring that individual responses could not be traced back to any participant. These measures were implemented to foster trust and ensure that participants felt safe and secure while contributing to the research.

## Data management and data analysis

Data management and analysis followed a systematic approach to ensure accuracy and reliability. Upon collection, questionnaires were thoroughly checked for completeness. Any incomplete questionnaires were discarded to maintain the integrity of the data.

The completed questionnaires were securely stored in a locked cupboard, accessible only to the principal investigator, until the data collection process was finalized. Once data collection was complete, an Excel spreadsheet generated from Kobo Toolbox was imported into SPSS version 26 for analysis. Additionally, a soft copy of the spreadsheet was saved on a flash drive and sent via email for backup purposes.

Data were analyzed using the Statistical Package for the Social Sciences (SPSS), version 26, with results presented in tables and charts. Continuous variables, such as age, were described using summary statistics, including means and standard deviations. Categorical variables, such as educational level and marital status, were summarized using frequency tables.

To assess caregivers' knowledge, toward the malaria vaccine, a scoring system was implemented. Each of the knowledge and attitudes sections of the questionnaire contained seven questions, with a maximum obtainable score of 7 for knowledge and 21 for attitudes.. For knowledge, correct answers received a score of one (1), while incorrect answers received zero (0). The total score for each participant was calculated based on their responses. A 60% cut-off point was used to classify adequate knowledge and positive attitudes, consistent with thresholds commonly applied in KAP (Knowledge, Attitudes, and Practices) studies. The 60% of the maximum obtainable score across all participants for each section was determined; those scoring at or above the 60% score were classified as having good knowledge, or positive attitudes while those scoring below the average were classified as having poor knowledge, or negative attitudes.

The anonymized dataset used for this study is provided in the supporting information (S1 Data).

## Results

### Socio-demographic characteristics

The mean age of the 410 caregivers was 33.6 and the standard deviation was 8.9. A total of 219 (53.4%) of caregivers were within the age group 21–35 years and 341 (83.2%) of them were females. Secondary education was the dominant educational level 160 (39.0%) and 340 (82.9%) were Christians. A majority 249 (60.7%) were married and 158 (38.5%) were self-employed. A vast majority 329 (80.2%) of the caregivers were direct parents of the under-five children and 391 (95.4%) were non-smokers (Table 2)

### Knowledge of caregivers of under five children on the malaria vaccine in the Tiko Health District

Regarding the knowledge of caregivers on the malaria vaccine (Table 3), 145 (35.4%) sources of information on the vaccine was healthcare worker and 249 (60.7%) were aware of the existence of the malaria vaccine. Of the 249 who were aware of the existence of the malaria vaccine, 145 (58.2%) did not know that the vaccine was approved in Cameroon and 109 (43.8%) knew that the purpose of the malaria vaccine was to boost immunity in order to prevent the malaria infection. Note: Because the study population included children aged 0–5 years, not all children were old enough to receive all four RTS,S vaccine doses. To avoid misclassification, we considered age-eligibility for each dose when interpreting the 'vaccine doses taken' variable. Children who had not yet reached the recommended age for Dose 2, Dose 3, or Dose 4 were classified as 'not yet age-eligible' rather than 'unvaccinated'. The categories '1 dose', '2 doses', '3 doses', and '4 doses' therefore represent only children who were age-eligible for those doses at the time of data collection.

Regarding caregivers' knowledge of the side effects of the malaria vaccine, 74 (31.8%) reported not knowing any side effect (Fig 1).

With respect to the overall knowledge of caregivers on the malaria vaccine, 60% of the maximum obtainable score was used as the cut off point for good knowledge, the overall good knowledge of the malaria vaccine was 26.6%.

### Attitudes of under-five caregivers toward the malaria vaccine in the Tiko Health District

Regarding the attitudes of caregivers toward the malaria vaccine (Table 4), 249 (60.7%) agreed that, vaccinating their children against malaria was crucial and 235 (57.3%) agreed that malaria vaccine was a safe option for protecting their children from malaria. A total of 219 (53.4) agreed that they trust the information provided by healthcare workers on the malaria vaccine and 169 (41.2%) agreed that completing the full course of the vaccine was essential for ensuring its effectiveness.

For the overall attitudes of caregivers toward the malaria vaccine (Table 4), a total of seven questions were asked with the options ranging from strongly disagree (0) to strongly agree (3). The maximum obtainable score was 21. A 60% cutoff point was used for overall positive attitudes. Those who scored 13 out of the 21 maximum obtainable score were

PLOS Global Public Health

**Table 2. Socio-demographic characteristics of caregivers (n = 410).**

| Variable | Category | Frequency | Percentage |
|---|---|---|---|
| Age group (years) | <21 | 20 | 4.8 |
| | 21-35 | 219 | 53.4 |
| | 36-50 | 149 | 36.3 |
| | 51-65 | 22 | 5.4 |
| Sex | Male | 69 | 16.8 |
| | Female | 341 | 83.2 |
| Education | No formal | 71 | 17.3 |
| | Primary | 81 | 19.8 |
| | Secondary | 160 | 39.0 |
| | Tertiary | 98 | 23.9 |
| Religion | Muslim | 70 | 17.1 |
| | Christian | 340 | 82.9 |
| Marital status | Single | 134 | 32.7 |
| | Married | 249 | 60.7 |
| | Widowed | 23 | 5.6 |
| | Divorced | 4 | 1.0 |
| Employment status | Student | 49 | 12.0 |
| | Unemployed | 83 | 20.2 |
| | Self-employed | 158 | 38.5 |
| | Employed | 120 | 29.3 |
| Relation with child | Not related | 22 | 5.4 |
| | Other relative | 36 | 8.8 |
| | Parents | 329 | 80.2 |
| | Sibling | 23 | 5.6 |
| Number of children | 1-2 | 236 | 57.6 |
| | 3-4 | 131 | 32.0 |
| | >4 | 43 | 10.5 |
| Household income (XAF, [1USD = 550.76XAF]) | <50000 | 138 | 33.7 |
| | 51000-100000 | 206 | 50.2 |
| | >100000 | 66 | 16.1 |
| Smoking status | Smoke | 19 | 4.6 |
| | Not smoke | 391 | 95.4 |
| Alcohol consumption | No | 127 | 31.0 |
| | Yes | 283 | 69.0 |
| Accessible health services | No | 64 | 15.6 |
| | Yes | 346 | 84.4 |
| Child ever had malaria before | No | 61 | 14.9 |
| | Yes | 349 | 85.1 |
| Trust health workers | No | 95 | 23.2 |
| | Yes | 315 | 76.8 |
| Go for general routine vaccination | No | 136 | 33.2 |
| | Yes | 274 | 66.8 |

**Table 3. Knowledge of under-five caregivers on the malaria vaccine in the Tiko Health District (n = 410).**

| Variable | Category | Frequency | Percentage |
|---|---|---|---|
| Source of information | Family and friends | 65 | 15.9 |
| | Health care provider | 145 | 35.4 |
| | None | 140 | 34.1 |
| | Others | 4 | 1.0 |
| | Social media | 56 | 13.7 |
| Aware of malaria vaccine | No | 161 | 39.3 |
| | Yes | 249 | 60.7 |
| Know that malaria vaccine is approved in Cameroon | No | 145 | 58.2 |
| | Yes | 104 | 41.8 |
| Purpose of the malaria vaccine | I do not know | 10 | 4.0 |
| | To boost the child immune. | 63 | 25.3 |
| | To prevent malaria infections. | 109 | 43.8 |
| | To reduce fever associated with malaria | 27 | 10.8 |
| | To treat malaria symptoms. | 40 | 16.1 |
| Doses of malaria vaccine Required age to start vaccinating children against malaria | 1 dose | 59 | 23.6 |
| | 2 doses | 64 | 25.7 |
| | 3 doses | 55 | 22.1 |
| | 4 doses | 30 | 12.0 |
| | I do not know | 41 | 16.5 |
| | At 1 year | 53 | 21.2 |
| | At 3 months | 49 | 19.7 |
| | At 6 months | 75 | 30.1 |
| | At birth | 33 | 13.3 |
| | I do not know | 39 | 15.7 |
| Malaria vaccine is 100% effective What you do when your child misses a vaccine dose/schedule | False | 165 | 66.3 |
| | True | 84 | 33.7 |
| | Consult a healthcare provider for advice | 89 | 35.7 |
| | Skip that dose and continue with the next one. | 42 | 16.9 |
| | Start the vaccination schedule again from the beginning | 79 | 31.7 |
| | Wait until the next schedule dose | 39 | 15.7 |

[1]Dose counts represent only children who were age-eligible for each respective RTS,S malaria vaccine dose at the time of data collection. Children who had not yet reached the recommended age for Dose 2, Dose 3, or Dose 4 were classified as 'not yet age-eligible' rather than 'unvaccinated.

classified as having overall positive attitudes and those who scored below 13 as having overall negative attitudes. Following this, 103 (25.1%) had overall positive attitudes toward the malaria vaccine.

**Factors associated with awareness of the malaria vaccine among caregivers of children 0–5 years**

At the level of the bivariable analysis using simple logistic regression with unadjusted/crude odd ratios (COR), four factors were found significantly associated with malaria vaccine awareness among caregivers of children 0–5 years. Factors with a p value of <0.2 were taken to the multivariable analysis to identify factors independently associated with malaria vaccine awareness using multiple logistic regression with adjusted odd rations.

Factors found significantly associated in the bivariable analysis were sex of caregiver, smoking status, trust in health workers, and going for general routine vaccination (Table 5).

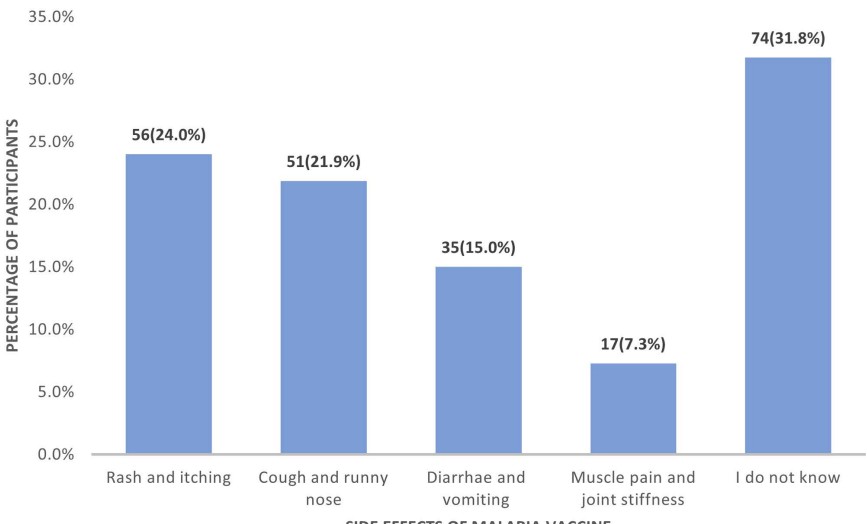

**Fig 1. Distribution of caregivers according to their knowledge of the side effects of the malaria vaccine.**

**Table 4. Attitudes of under-five caregivers toward the malaria vaccine.**

| Variable | Strongly disagree | Disagree | Agree | Strongly agree |
|---|---|---|---|---|
| I believe that vaccinating my child against malaria is crucial for their health | 21(5.1) | 68(16.6) | 249(60.7) | 72(17.6) |
| Malaria vaccine is a safe option for protecting my child from malaria. | 100(24.4) | 54(13.2) | 235(57.3) | 21(5.1) |
| Trust the information provided by healthcare professionals. | 24(5.9) | 137(33.4) | 219(53.4) | 30(7.3) |
| Completing the full course of malaria vaccination is essential for ensuring its effectiveness. | 130(31.7) | 67(16.3) | 169(41.2) | 44(10.7) |
| Believe that the benefits of the malaria vaccine outweigh any potential risks | 134(32.7) | 78(19.0) | 166(40.5) | 32(7.8) |
| My child should receive the malaria vaccine according to the recommended vaccination schedule.ended vaccination schedule | 126(30.7) | 81(19.8) | 161(39.3) | 42(10.2) |
| I believe that the malaria vaccine is an important tool in preventing malaria related illnesses in children. | 33(8.0) | 174(42.4) | 174(42.4) | 29(7.1) |
| **Overall attitudes** | **Positive = 103 (25.1%)** | | | |
| | **Negative = 307 (74.9%)** | | | |

After adjusting for potential confounders in the multivariable logistic regression model, three factors remained independently associated with caregivers' awareness of the malaria vaccine (Table 6). Caregivers whose children had previously experienced malaria were significantly less likely to be aware of the malaria vaccine compared with those whose children had no prior malaria history (Adjusted Odds Ratio [AOR] = 0.55; 95% Confidence Interval [CI]: 0.28–0.97; p = 0.040).

In contrast, caregivers who reported trusting health workers were significantly more likely to be aware of the malaria vaccine than those who did not trust health workers (AOR = 3.02; 95% CI: 1.83–4.99; p < 0.001). Similarly, caregivers who routinely took their children for general childhood immunization were more likely to be aware of the malaria vaccine compared with those who did not attend routine immunization services (AOR = 3.57; 95% CI: 2.27–5.60; p < 0.001).

The multivariable logistic regression model demonstrated adequate fit to the data (Hosmer–Lemeshow p = 0.670) and explained approximately 29.9% of the variance in caregivers' awareness of the malaria vaccine (Pseudo $R^2$ = 0.299).

**Table 5. Factors associated with awareness of the malaria vaccine using simple logistic regression.**

| Variable | Category | Aware of malaria vaccine | | COR | 95% CI for COR | | p value |
|---|---|---|---|---|---|---|---|
| | | No | Yes | | Lower | Upper | |
| Sex | Female | 124(30.2) | 217(52.9) | 2.02 | 1.20 | 3.41 | 0.008 |
| | Male | 37(9) | 32(7.8) | 1 | | | |
| Education | Tertiary | 30(7.3) | 68(16.6) | 1.48 | 0.78 | 2.80 | 0.234 |
| | Secondary | 63(15.4) | 97(23.7) | 1.00 | 0.57 | 1.78 | 0.993 |
| | Primary | 40(9.8) | 41(10) | 0.67 | 0.35 | 1.27 | 0.219 |
| | No formal | 28(6.8) | 43(10.5) | 1 | | | |
| Religion | Muslim | 129(31.5) | 211(51.5) | 1.38 | 0.82 | 2.31 | 0.226 |
| | Christian | 32(7.8) | 38(9.3) | 1 | | | |
| Marital status | Widowed | 2(0.5) | 2(0.5) | 0.76 | 0.10 | 5.58 | 0.790 |
| | Divorced | 9(2.2) | 14(3.4) | 1.19 | 0.48 | 2.93 | 0.710 |
| | Married | 92(22.4) | 157(38.3) | 1.30 | 0.85 | 2.00 | 0.226 |
| | Single | 58(14.1) | 76(18.5) | 1 | | | |
| Health related work/field of study | Yes | 10(2.4) | 23(5.6) | 1.54 | 0.71 | 3.32 | 0.274 |
| | No | 151(36.8) | 226(55.1) | 1 | | | |
| Relationship with child | Siblings | 11(2.7) | 12(2.9) | 1.09 | 0.34 | 3.51 | 0.884 |
| | Parents | 124(30.2) | 205(50) | 1.65 | 0.70 | 3.93 | 0.255 |
| | Others | 15(3.7) | 21(5.1) | 1.40 | 0.48 | 4.07 | 0.536 |
| | No direct relation | 11(2.7) | 11(2.7) | 1 | | | |
| Smoking status | Smoker | 149(36.3) | 242(59) | 2.78 | 1.07 | 7.23 | 0.035 |
| | Non-smoker | 12(2.9) | 7(1.7) | 1 | | | |
| Alcohol consumption | Yes | 114(27.8) | 169(41.2) | 0.87 | 0.57 | 1.34 | 0.530 |
| | No | 47(11.5) | 80(19.5) | 1 | | | |
| Accessible health services | Yes | 130(31.7) | 216(52.7) | 1.56 | 0.91 | 2.67 | 0.104 |
| | No | 31(7.6) | 33(8) | 1 | | | |
| Child has previous malaria infection | Yes | 144(35.1) | 205(50) | 0.55 | 0.30 | 1.00 | 0.050 |
| | No | 17(4.1) | 44(10.7) | 1 | | | |
| Trust health workers | Yes | 103(25.1) | 212(51.7) | 3.23 | 2.01 | 5.19 | <0.001 |
| | No | 58(14.1) | 37(9) | 1 | | | |
| Go for general routine vaccination | Yes | 80(19.5) | 194(47.3) | 3.57 | 2.32 | 5.49 | <0.001 |
| | No | 81(19.8) | 55(13.4) | 1 | | | |

*COR: Crude Odd Ratio, CI: Confidence Interval*

## Discussion

This study assessed caregivers' knowledge, attitudes, and factors associated with awareness of the malaria vaccine among caregivers of children under five years of age in the Tiko Health District, Cameroon. Overall, the findings reveal low levels of both knowledge and positive attitudes toward the malaria vaccine, despite moderate awareness of its existence. Guided by the Health Belief Model (HBM), these findings highlight important cognitive and contextual determinants of malaria vaccine awareness that are relevant for public health programming.

Only 26.6% of caregivers demonstrated good overall knowledge about the malaria vaccine. Although 60.7% of respondents had heard about the vaccine, deeper understanding was limited. A significant proportion of those aware of the vaccine did not know it had been approved in Cameroon, and many had misconceptions about its purpose. This low knowledge level is concerning as studies have shown that, misconception as a result of inadequate knowledge is

**Table 6. Factors independently associated with the malaria vaccine awareness among caregivers.**

| Variable | Category | Aware of malaria vaccine | | AOR | 95% CI for COR | | p value |
| --- | --- | --- | --- | --- | --- | --- | --- |
| | | No | Yes | | Lower | Upper | |
| Religion | Muslim | 129(31.5) | 211(51.5) | 1.44 | 0.79 | 2.64 | 0.237 |
| | Christian | 32(7.8) | 38(9.3) | 1 | | | |
| Smoking status | Smoker | 149(36.3) | 242(59) | 1.96 | 0.69 | 5.54 | 0.204 |
| | Non-smoker | 12(2.9) | 7(1.7) | 1 | | | |
| Alcohol consumption | Yes | 114(27.8) | 169(41.2) | 0.73 | 0.44 | 1.20 | 0.21 |
| | No | 47(11.5) | 80(19.5) | 1 | | | |
| Child has previous malaria infection | Yes | 144(35.1) | 205(50) | 0.55 | 0.28 | 0.97 | **0.040** |
| | No | 17(4.1) | 44(10.7) | 1 | | | |
| Trust health workers | Yes | 103(25.1) | 212(51.7) | 3.02 | 1.83 | 4.99 | **<0.001** |
| | No | 58(14.1) | 37(9) | 1 | | | |
| Go for general routine vaccination | Yes | 80(19.5) | 194(47.3) | 3.57 | 2.27 | 5.60 | **<0.001** |
| | No | 81(19.8) | 55(13.4) | 1 | | | |

Abbreviations: COR, Crude Odds Ratio; AOR, Adjusted Odds Ratio; CI, Confidence Interval.

Model statistics: Number of observations = 410; Pseudo $R^2$ = 0.299; Hosmer–Lemeshow goodness-of-fit test p = 0.670.

negatively associated with the malaria vaccine uptake [27,28].Within the HBM framework, inadequate knowledge may reduce perceived benefits of vaccination and increase perceived barriers, thereby undermining caregivers' readiness to engage in preventive behaviors [26]. This is consistent with previous studies conducted in various regions of sub-Saharan Africa like the one in Nigeria and Tanzania [29,30], which often report low levels of knowledge among caregivers regarding malaria vaccination. For instance, a study conducted in Tanzania in 2025 reported 14.7% awareness of caregivers on the malaria vaccine [27]. Another study in Nigeria also reported a 30% awareness of the malaria vaccine reported in Northern Nigeria [31]. Although some studies reported slightly higher awareness levels (around 30–40%), these differences are modest and may reflect contextual variations rather than major epidemiological differences [32].

Attitudinal assessment revealed that only 25.1% of caregivers had a positive attitude toward the malaria vaccine. Although a majority (60.7%) agreed that vaccinating children is crucial, concerns about safety, efficacy, and mistrust of health information remain high. Less than half of caregivers agreed that completing the full vaccine course is essential, and over 30% expressed skepticism about the vaccine's safety. From an HBM perspective, these concerns reflect high perceived barriers and insufficient perceived benefits, which may explain the observed gap between general support for vaccination and willingness to fully embrace the malaria vaccine [26].

These findings mirror results from Tanzania, where caregivers expressed concerns about vaccine side effects and effectiveness, which led to vaccine hesitancy [33]. Similarly, in Uganda, positive attitudes were reported only when caregivers had received direct counseling and community engagement interventions [34]. The relatively low trust in information from health professionals in our study (53.4% agreeing or strongly agreeing) further highlights the need for improved communication strategies that foster trust and transparency.

The low proportion of caregivers who exhibited both good knowledge and positive attitudes is particularly troubling. It suggests that many caregivers, even if aware of the vaccine, may remain hesitant due to unresolved doubts or misinformation. This disconnects between awareness and confidence can undermine vaccine rollout efforts and delay progress in malaria control.

In this study, we identified key factors associated with under-five caregivers' awareness of the malaria vaccine. Caregivers whose children had previously experienced malaria were less likely to be aware of the malaria vaccine. This finding may reflect a reliance on curative care following malaria episodes, with less attention given to preventive strategies. Within the HBM framework, prior malaria experience may heighten perceived severity but does not necessarily translate into

increased perceived benefits of vaccination, particularly in the absence of effective counseling during treatment encounters [26]. This finding may reflect a perception among affected caregivers that treatment is the primary recourse, thereby reducing attention to preventive measures such as vaccination [35]. It could also suggest missed opportunities for health education during malaria treatment encounters, where health personnel could leverage the moment to introduce and promote the malaria vaccine.

Consistent with the HBM, trust in health workers emerged as a strong positive predictor of caregivers' awareness of the malaria vaccine. Trust in healthcare providers likely functions as a key cue to action, facilitating the acceptance of new health information and innovations. Caregivers who expressed trust in health professionals were more likely to be aware of the malaria vaccine than those who did not. This underscores the critical role of trust in health communication and uptake of health innovations. It suggests that enhancing the credibility and approachability of health workers may significantly improve public receptiveness to new health interventions, including vaccines [36].

Furthermore, participation in routine childhood immunization was strongly associated with awareness of the malaria vaccine. Caregivers who took their children for general vaccination were significantly more likely to be aware of the malaria vaccine than those who did not. This likely reflects greater engagement with the health system, increased exposure to health promotion messages, and habitual preventive health-seeking behaviors among these caregivers. These findings are similar to what was reported by a study in Northern Nigeria where prior infection of malaria and prior experience with vaccination were significantly associated with the malaria vaccine awareness [37].

Together, these findings highlight the importance of integrating malaria vaccine awareness into routine health service touchpoints, particularly childhood immunization programs, and underscore the need to build and maintain trust between communities and health providers. Tailored health education strategies targeting caregivers of children with prior malaria episodes may also be necessary to close the awareness gap and maximize uptake of the malaria vaccine.Top of Form-Bottom of Form.

These results highlight the need for intensified community sensitization efforts and caregiver-focused health education campaigns. Interventions should not only provide factual information about the vaccine but also address cultural beliefs, fears, and misinformation.

Training healthcare workers in interpersonal communication and community engagement is also essential. Given that health personnel are the most trusted sources of information, enhancing their capacity to deliver clear, empathetic, and persuasive vaccine messages could significantly influence caregiver perceptions and behaviors.

## Strengths and limitations

A key strength of this study is its community-based approach, which allowed for the inclusion of a diverse and representative sample of caregivers of under-five children across the Tiko Health District. The use of trained research assistants and a standardized structured questionnaire helped ensure consistency in data collection and minimized interviewer bias. Additionally, the study addresses an important gap in the literature on malaria vaccine awareness in a real-world setting, providing timely and relevant evidence to inform public health interventions in Cameroon.

However, the study has some limitations. Its cross-sectional design limits the ability to infer causality between the identified factors and caregivers' awareness of the malaria vaccine. Self-reported data may also be subject to recall and social desirability bias, particularly for sensitive topics such as trust in health workers or participation in vaccination programs. Also, while our overall sample size was adequate and determined using standard methods, we acknowledge that the number of events per predictor variable (EPV) in the multivariable model may have been limited for certain covariates.

## Conclusion

In conclusion, this study reveals low levels of both knowledge and positive attitudes toward the malaria vaccine among caregivers of under-five children in the Tiko Health District. These gaps pose a significant barrier to successful vaccine

implementation and require urgent, targeted interventions. Strengthening community health education, improving trust in health systems, and engaging caregivers directly will be crucial in enhancing the acceptance and uptake of the malaria vaccine in this high-risk population. Factors found associated with caregivers' awareness of the malaria vaccine were trust in health workers, going for general child routine vaccination and history of malaria infection in the child.

## Supporting information

**S1 Data. Anonymized dataset used in the study.**
(XLSX)

## Author contributions

**Conceptualization:** Maureen Abiache Idang, Divine Nsobinenyui, Chrisantus Eweh Ukah, Larissa Kumenyuy Yunika, Claudia Ngeha Ngu, Randolf Fuanghene Wefuan, Ndip Esther Ndip, Mirabelle Pandong Feguem, Dickson Shey Nsagha.

**Data curation:** Maureen Abiache Idang, Divine Nsobinenyui, Chrisantus Eweh Ukah, Larissa Kumenyuy Yunika, Claudia Ngeha Ngu, Syveline Zuh Dang, Mirabelle Pandong Feguem, Dickson Shey Nsagha.

**Formal analysis:** Chrisantus Eweh Ukah, Larissa Kumenyuy Yunika, Ndip Esther Ndip.

**Investigation:** Maureen Abiache Idang, Divine Nsobinenyui, Chrisantus Eweh Ukah, Claudia Ngeha Ngu, Randolf Fuanghene Wefuan, Syveline Zuh Dang, Mirabelle Pandong Feguem.

**Methodology:** Maureen Abiache Idang, Divine Nsobinenyui, Chrisantus Eweh Ukah, Larissa Kumenyuy Yunika, Claudia Ngeha Ngu, Randolf Fuanghene Wefuan, Mirabelle Pandong Feguem, Dickson Shey Nsagha.

**Project administration:** Divine Nsobinenyui, Chrisantus Eweh Ukah.

**Supervision:** Divine Nsobinenyui, Chrisantus Eweh Ukah, Dickson Shey Nsagha.

**Validation:** Divine Nsobinenyui, Chrisantus Eweh Ukah, Dickson Shey Nsagha.

**Visualization:** Chrisantus Eweh Ukah, Larissa Kumenyuy Yunika, Ndip Esther Ndip.

**Writing – original draft:** Maureen Abiache Idang, Chrisantus Eweh Ukah, Larissa Kumenyuy Yunika, Randolf Fuanghene Wefuan, Syveline Zuh Dang.

**Writing – review & editing:** Maureen Abiache Idang, Divine Nsobinenyui, Chrisantus Eweh Ukah, Larissa Kumenyuy Yunika, Randolf Fuanghene Wefuan, Ndip Esther Ndip, Dickson Shey Nsagha.

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
