## [Decision Letter · Decision Letter 0]

8 Jul 2025

PGPH-D-25-01040

Knowledge and attitudes of caregivers of under-five children toward the malaria vaccine in the Tiko Health District, Cameroon: A community-based cross-sectional study

Dear Dr. Chrisantus Eweh Ukah,

Thank you for submitting your manuscript to PLOS Global Public Health. After careful consideration, we feel that it has merit but does not fully meet PLOS Global Public Health’s publication criteria as it currently stands. Therefore, we invite you to submit a revised version of the manuscript that addresses the points raised during the review process.

EDITOR: Please address reviewers' comments, including the ones in the attached documents.

Please ensure that your decision is justified on PLOS Global Public Health’s publication criteria  and not, for example, on novelty or perceived impact.

Please submit your revised manuscript by **22/08/2025.** If you will need more time than this to complete your revisions, please reply to this message or contact the journal office at globalpubhealth@plos.org. Please include the following items when submitting your revised manuscript:

We look forward to receiving your revised manuscript.

Kind regards,

Peter Bai James, PhD

Academic Editor

Journal Requirements:

1. In the online submission form, you indicated that [The dataset supporting the conclusions of this article is available from the corresponding author upon reasonable request.].

a. In a public repository,

b. Within the manuscript itself, or

c. Uploaded as supplementary information.

Additional Editor Comments (if provided):

Reviewers' comments:

Reviewer's Responses to Questions

**Comments to the Author**

1. Does this manuscript meet PLOS Global Public Health’s publication criteria?

Reviewer #1: No

Reviewer #2: Partly

2. Has the statistical analysis been performed appropriately and rigorously?

Reviewer #1: No

Reviewer #2: No

3. Have the authors made all data underlying the findings in their manuscript fully available (please refer to the Data Availability Statement at the start of the manuscript PDF file)?

Reviewer #1: No

Reviewer #2: No

4. Is the manuscript presented in an intelligible fashion and written in standard English?

Reviewer #1: Yes

Reviewer #2: Yes

Reviewer #1: Based on my review, the paper requires major revisions before acceptance for publication. The study addresses an important public health problem which is the question of knowledge and attitude or caregivers towards the new malaria vaccine. The potential public health impact of these findings is high, making it crucial that the study is presented with utmost clarity and rigor. I have attached my comments to the authors.

Reviewer #2: Abstract

“A descriptive cross-sectional survey”. Please rephrase as “A cross-sectional survey”

Background

1. Check and correct references in line 60 and 67

2. “Moreover, caregivers' socio-economic 80 status, education level, and exposure to health education initiatives can further affect their understanding and acceptance of new interventions like the malaria vaccine”. Provide a reference

3. While the authors have demonstrated the importance of understanding caregivers knowledge and attitudes toward the malaria vaccine, the research gap that authors are seeking to close, is not clearly established. Has there been similar studies in the study context? Are there similar studies in other contexts in Cameroon or outside Cameroon but from the Sub-Saharan African region? What are the findings of those studies, and the research gap that need to be addressed. Overall, authors should consider delving into the literature of what has been done so far either within the study context or outside of the study context, and then highlight what new thin the current study is adding to the existing literature.

4. There is currently no theoretical underpinning for this study. A study like this must be rooted in in a relevant theoretical framework such as the theory of planned behaviours. While it is not compulsory to use the proposed theory, it is imperative to have the study guided by a relevant theory.

Materials And Methods

1. At line 90, please remove the word “descriptive”. Study design can be cross-sectional or longitudinal but not descriptive. Descriptive is part of the analytical approach.

2. “This study was conducted in some selected health areas of the Tiko Health District”. Please state the selected health areas.

3. The “ETHICAL CONSIDERATIONS” should follow immediately after the “Data Collection” section.

Data Analysis

• A sample size over 400 is large enough for test of association. While the authors did a good job with descriptives, that is too basic/not robust and cannot be used to confirm any hypothesis about knowledge and attitudes towards malaria vaccine. I strongly recommend a more advanced analysis/test of association for statistical significance. From the Tables, I can tell the variable “Aware of malaria vaccine”(outcome measure) is binary, hence a multivariable logistic regression model is more appropriate. See one the pioneering study on child malaria vaccine for guidance: https://doi.org/10.1371/journal.pone.0296934

Results

Kindly remove the “Totals” from Table 1 and 2.

Discussion

The revised version of the manuscript should discuss significant independent variables/predictors form the regression model(s), and discussions should be guided by the theoretical framework of the study as well.

Additional Comments: The study have merit. However, the methodology/analytical approach is currently weak and needs major revision for robustness.

**Do you want your identity to be public for this peer review?** For information about this choice, including consent withdrawal, please see our Privacy Policy

Reviewer #1: **Yes:** Suleiman Idris Ahmad

Reviewer #2: No

---

## [Decision Letter · Decision Letter 1]

27 Nov 2025

PGPH-D-25-01040R1

Knowledge, attitudes and factors associated with the awareness of caregivers of under-five children regarding the malaria vaccine in the Tiko Health District, Cameroon: A community-based cross-sectional study

Dear Dr. Ukah,

Thank you for submitting your manuscript to PLOS Global Public Health. After careful consideration, we feel that it has merit but does not fully meet PLOS Global Public Health’s publication criteria as it currently stands. Therefore, we invite you to submit a revised version of the manuscript that addresses the points raised during the review process.

We look forward to receiving your revised manuscript.

Kind regards,

Helen Howard

Staff Editor

Journal Requirements:

Additional Editor Comments (if provided):

Reviewers' comments:

Reviewer's Responses to Questions

**Comments to the Author**

Reviewer #2: (No Response)

publication criteria?

Reviewer #2: (No Response)

3. Has the statistical analysis been performed appropriately and rigorously?

Reviewer #2: (No Response)

4. Have the authors made all data underlying the findings in their manuscript fully available (please refer to the Data Availability Statement at the start of the manuscript PDF file)?

Reviewer #2: (No Response)

5. Is the manuscript presented in an intelligible fashion and written in standard English?

Reviewer #2: (No Response)

Reviewer #2: Kindly see the attached word document with comments.

**Do you want your identity to be public for this peer review?** For information about this choice, including consent withdrawal, please see our Privacy Policy

Reviewer #2: No

---

## [Decision Letter · Decision Letter 2]

20 Jan 2026

PGPH-D-25-01040R2

Knowledge, attitudes and factors associated with the awareness of caregivers of under-five children regarding the malaria vaccine in the Tiko Health District, Cameroon: A community-based cross-sectional study

Dear Dr. Ukah,

Thank you for submitting your manuscript to PLOS Global Public Health. After careful consideration, we feel that it has merit but does not fully meet PLOS Global Public Health’s publication criteria as it currently stands. Therefore, we invite you to submit a revised version of the manuscript that addresses the points raised during the review process.

We look forward to receiving your revised manuscript.

Kind regards,

Helen Howard

Staff Editor

Journal Requirements:

Additional Editor Comments (if provided):

Reviewers' comments:

Reviewer's Responses to Questions

**Comments to the Author**

Reviewer #2: All comments have been addressed

Reviewer #3: (No Response)

publication criteria?

Reviewer #2: Yes

Reviewer #3: Yes

3. Has the statistical analysis been performed appropriately and rigorously?

Reviewer #2: Yes

Reviewer #3: Yes

4. Have the authors made all data underlying the findings in their manuscript fully available (please refer to the Data Availability Statement at the start of the manuscript PDF file)?

Reviewer #2: Yes

Reviewer #3: Yes

5. Is the manuscript presented in an intelligible fashion and written in standard English?

Reviewer #2: Yes

Reviewer #3: Yes

Reviewer #2: The authors have adopted a theoretical framework and also situated discussions with the framework. Thus, I recommend that the manuscript be accepted but wit thorough proofreading from authors prior to publication.

Reviewer #3: Dear authors, thank you for the opportunity to read your work.

While the study's significant results mostly reinforce what is already known about the situation of the malaria vaccine uptake in malaria-endemic regions, I think it is important to report results from multiple data sources and areas.

The following points would improve the manuscript.

Abstract:

1. L16-17, “The introduction of the RTS,S/AS01 (Mosquirix) malaria vaccine offers hope”, you should also mention R21, even though it has not introduced in Cameroon. It should be a general topic regarding malaria vaccines in this introduction part, and it should be comprehensive.

2. L24, you do not need to mention SPSS and its version in the Abstract; it is not so important to be included here.

3. L26, What is the scoring system?

4. L30, I think the statistics of age should be summarized as median and IQR instead of means and SD.

5. You do not need to include the p-value because you included the 95% CI; it can reduce the word count of the abstract.

6. L39-41, These are the results of your analysis, and it is a repetition of what you stated above. Instead, you need to write a conclusion from your study here.

Materials and methods:

7. L126-127: The date information was repeated.

8. L133-134: What is the proportion? I guess it is the proportion of caregivers' awareness of malaria vaccine or something? It must be the primary interest of the study.

9. L140: You need to explain why it was made up to 410.

10. L178: You wrote four distinct sections, but it seems there are only three sections.

11. L238: SPSS should be spelled out when it appears first, such as L235.

12. L242: There are no pie charts, and I do not even think you need pie charts. The table is enough.

13. L248: Can you explain why you set the cutoff of 60%?

Results:

14. L258: Is “under children” a typo?

15. Table 1: Category for number of children should be >4 instead of 4+. The category for household income should be >100,000 instead of 100000+. And what is the currency, and how much is it in USD?

16. Table 2: For the doses of malaria vaccine, because children were 0-5 years, I guess some small children have not reached the age of Dose 4 yet. Did you consider such age eligibility for each child and each dose?

17. I don’t think Figure 1 is necessary. It can be a table.

18. L279-281, the statements of “With respect to the overall knowledge of caregivers on the malaria vaccine, 60% of the maximum obtainable score was used as the cut off point for good knowledge. A total of seven questions were asked with a maximum obtainable score of 7.” Is repetition.

Discussion:

19. L338-339, Not clear how guided by the HBM. You should mention the details of the model in the Methods section.

20. L354-355, I do not think the difference between 30% and 40% is big enough, especially since you did not include confidence intervals at all in these figures.

21. I think the sample size is not enough for the logistic regression. For example, the factors of female, higher education, Muslim, and accessible health services seemed to have a higher tendency to be aware of malaria vaccines, according to Table 4. You should mention these things, and also, the sample size should be mentioned as one of the limitations of your study.

**Do you want your identity to be public for this peer review?** For information about this choice, including consent withdrawal, please see our Privacy Policy

Reviewer #2: No

Reviewer #3: **Yes:** Yura K Ko

---

## [Decision Letter · Decision Letter 3]

10 Feb 2026

PGPH-D-25-01040R3

Knowledge, attitudes and factors associated with the awareness of caregivers of under-five children regarding the malaria vaccine in the Tiko Health District, Cameroon: A community-based cross-sectional study

Dear Dr. Ukah,

Thank you for submitting your manuscript to PLOS Global Public Health. After careful consideration, we feel that it has merit but does not fully meet PLOS Global Public Health’s publication criteria as it currently stands. Therefore, we invite you to submit a revised version of the manuscript that addresses the points raised during the review process.

All previous reviewer comments have been addressed, however I have one final recommendation from an editorial perspective to enhance interpretation of your results. Please consider including a sentence or two in the introduction section explaining when RTS,S was approved for use in Cameroon, and when the vaccine was first available to children within the study district. This information will help to contextualise the reported findings, and make it easier for readers to compare these levels of knowledge and awareness across other settings as vaccine scale-up continues.

We look forward to receiving your revised manuscript.

Kind regards,

Ruth Ashton, Ph.D.

Academic Editor

Journal Requirements:

Additional Editor Comments (if provided):

Please consider including a sentence or two in the introduction section explaining when RTS,S was approved for use in Cameroon, and when the vaccine was first available to children within the study district. This information will help to contextualise the reported findings, and make it easier for readers to compare these levels of knowledge and awareness across other settings as vaccine scale-up continues.

Reviewers' comments:

Reviewer's Responses to Questions

**Comments to the Author**

Reviewer #3: All comments have been addressed

publication criteria?

Reviewer #3: Yes

3. Has the statistical analysis been performed appropriately and rigorously?

Reviewer #3: Yes

4. Have the authors made all data underlying the findings in their manuscript fully available (please refer to the Data Availability Statement at the start of the manuscript PDF file)?

Reviewer #3: Yes

5. Is the manuscript presented in an intelligible fashion and written in standard English?

Reviewer #3: Yes

Reviewer #3: I appreciate the authors’ careful and comprehensive responses, and I consider that all comments have been adequately addressed.

**Do you want your identity to be public for this peer review?** For information about this choice, including consent withdrawal, please see our Privacy Policy

Reviewer #3: **Yes:** Yura K Ko

---

## [Editor Report · Decision Letter 4]

13 Feb 2026

Knowledge, attitudes and factors associated with the awareness of caregivers of under-five children regarding the malaria vaccine in the Tiko Health District, Cameroon: A community-based cross-sectional study

PGPH-D-25-01040R4

Dear Mr Ukah,

We are pleased to inform you that your manuscript 'Knowledge, attitudes and factors associated with the awareness of caregivers of under-five children regarding the malaria vaccine in the Tiko Health District, Cameroon: A community-based cross-sectional study' has been provisionally accepted for publication in PLOS Global Public Health.

Best regards,

Ruth Ashton, Ph.D.

Academic Editor